# Safety of human-AI cooperative decision-making within intensive care: A physical simulation study

Paul Festor[1,2☺], Myura Nagendran[1,2,3☺], Anthony C. Gordon[1,3], Aldo A. Faisal [1,2,4*‡], Matthieu Komorowski[1,3‡]

1 UKRI Centre for Doctoral Training in AI for Healthcare, Imperial College London, London, United Kingdom, 2 Departments of Bioengineering and Computing, Brain and Behavior Lab, Imperial College London, London, United Kingdom, 3 Division of Anaesthetics, Pain Medicine and Intensive Care, Imperial College London, London, United Kingdom, 4 Digital Health and Data Science, Universität Bayreuth, Bayreuth, Germany

☺ These authors contributed equally to this work.
‡ These authors jointly supervised this work
* a.faisal@imperial.ac.uk

## Abstract

The safety of Artificial Intelligence (AI) systems is as much one of human decision-making as a technological question. In AI-driven decision support systems, particularly in high-stakes settings such as healthcare, ensuring the safety of human-AI interactions is paramount, given the potential risks of following erroneous AI recommendations. To explore this question, we ran a safety-focused clinician-AI interaction study in a physical simulation suite. Physicians were placed in a simulated intensive care ward, with a human nurse (played by an experimenter), an ICU data chart, a high-fidelity patient mannequin and an AI recommender system on a display. Clinicians were asked to prescribe two drugs for the simulated patients suffering from sepsis and wore eye-tracking glasses to allow us to assess where their gaze was directed. We recorded clinician treatment plans before and after they saw the AI treatment recommendations, which could be either 'safe' or 'unsafe'. 92% of clinicians rejected unsafe AI recommendations vs 29% of safe ones. Physicians paid increased attention (+37% gaze fixations) to unsafe AI recommendations vs safe ones. However, visual attention on AI explanations was not greater in unsafe scenarios. Similarly, clinical information (patient monitor, patient chart) did not receive more attention after an unsafe versus safe AI reveal suggesting that the physicians did not look back to these sources of information to investigate why the AI suggestion might be unsafe. Physicians were only successfully persuaded to change their dose by scripted comments from the bedside nurse 5% of the time. Our study emphasises the importance of human oversight in safety-critical AI and the value of evaluating human-AI systems in high-fidelity settings that more closely resemble real world practice.

## Author summary

Artificial intelligence (AI) demonstrates increasingly promising results for supporting medical decision-making. However, most systems are evaluated outside of the clinical

**Data availability statement:** The data and code (in the form of Jupyter notebooks) to reproduce the results and figures in both the manuscript and the supplementary appendices are available at: https://figshare.com/s/78c5ff5c6031f701c0d1

**Funding:** This work was funded by the University of York and the Lloyd's Register Foundation through the Assuring Autonomy International Programme (Project Reference 03/19/07) and supported by the National Institute for Health Research (NIHR) Imperial Biomedical Research Centre (BRC). PF and MN were supported by a PhD studentship of the UKRI Centre for Doctoral Training in AI for Healthcare (EP/S023283/1). ACG was supported by an NIHR Research Professorship (RP-2015-06-018). AAF was supported by a UKRI Turing AI Fellowship (EP/V025449/1). This study/project/report is independent research funded by the NIHR (Artificial Intelligence, 'Validation of a machine learning tool for optimal sepsis treatment', AI_AWARD01869). The funders had no role in study design, data collection and analysis, decision to publish, or preparation of the manuscript.

**Competing interests:** MK has received consulting fees from Philips Healthcare, and speaker honoraria from GE Healthcare. The other authors declare that there are no competing interests.

context within which they will be used, predominantly to avoid incurring risks for patients. When used at the bedside, these AI systems will be part of larger human-computer decision teams and little work evaluates the safety of this tandem as a whole. Here, we use a physical simulation suite and eye-tracking glasses to analyse the interaction between doctors and an AI support tool. Using the examplar of cardiovascular management in patients with sepsis, we report clinician behaviour when facing both (intentionally crafted) 'safe' and 'unsafe' recommendations from the AI system. We show that unsafe recommendations capture the attention of clinicians, increase the number of requests for a second opinion, and would be stopped over 90% of the time. However, unsafe AI recommendations did not draw more attention to accompanying explanations so the cognitive process behind the decision to reject them is not elucidated here. Working in a simulation suite also allowed us to study the potential human factor of decision safety: an experimenter playing the role of a nurse was able to convince 5% of clinicians to adopt an unsafe AI recommendation using pre-set challenge arguments. Combined with an objective measure of attention via eye-tracking, our approach allows us to capture and analyse human-AI interactions in safety-critical medical settings.

## Introduction

Artificial Intelligence (AI) driven systems have an increasingly prominent role in decision-making including in high-stakes (e.g., potentially life-threatening) settings [1–3]. However, the final decision usually remains in human hands. Therefore, understanding how AI recommendations impact end-user behaviour is crucial, and recent work has highlighted differences in the human perception of human vs AI advice such as the latter being significantly more influential on the final decision [4,5]. In high-risk shared decision-making settings, even non-autonomous AI-decision-support tools have high safety assurance requirements such as rigorous testing and usage auditing by notified bodies [6,7]. Understanding the human-AI cooperation dynamic is key to ensuring safe AI deployment. Work has been done to use human feedback as an active learning mechanism for robots, sometimes enforcing safe behaviour [8–10]. Yet, the evolution of AI safety has little focus on quantifying its impact on the final decision-maker, the human.

We study safety in human-AI cooperative settings in healthcare because healthcare reflects many of the issues that make real-world AI deployment challenging: regulatory requirements, high stakes, high cognitive load on decision-makers, and limited availability of human experts. Furthermore, the post-hoc rationalisations of human decision-makers can sometimes be "black-box" in nature, just as AI decisions can be [11]. In situations where the "optimal" decision has historically been unclear, and where recommending decisions could surpass the standard of care [12–14], the safety-assessment challenge becomes even harder. One such example is cardiovascular management during sepsis where the optimal personalised doses of intravenous (IV) fluids and vasopressors remain unknown [15,16]. Their personalisation has been designated as a research priority by the international reference taskforce the "Surviving Sepsis Campaign" [17]. Sepsis treatment in intensive care embodies all the characteristics that make healthcare AI both a difficult challenge but also one that could have a significant beneficial patient impact if successful. Sepsis affects up to 50 million people around the world with 20% annually dying. It represents the most common cause of death in the hospital and the most expensive condition treated in hospital [18]. As such, applying AI and data-driven approaches to deliver precision medicine therapy for sepsis is a global and urgent research priority. Yet, the translation to the bedside has been slow due to concerns for patient safety and clinician trust in these systems [19–21].

Attempts have been made to improve the safety profile of AI-driven decision support in retrospective intensive care settings [12–14]. Still, the necessity of prospective and higher fidelity evaluations involving clinical end users is clear from recent examples in other fields [13]. For instance, an acute kidney injury alert system showing good performance on retrospective data was found to worsen outcomes when deployed in a real-world setting, illustrating the need for a careful transition between retrospective testing and prospective deployment of digital systems [22]. Safely transitioning from "bytes to bedside" is a particularly complex challenge because of the dynamic interaction with human users who are prone to biases and can behave in unpredictable ways [23–26].

In response to the growing emphasis on ecologically valid experimentation, we need to evaluate the dynamics of human-AI cooperative systems in the same context [27,28], and therefore we run our behavioural experiment in a physical simulation suite. Simulation exercises have historically been used as a widely accepted training tool for both modelling high-fidelity situations and capturing patterns of human behaviour. They now form a core part of medical training [29]. This immersive approach enables physicians to respond to bedside stimuli more realistically, aligning their behaviours with actual clinical practice [30] Furthermore, this shift towards a more realistic setting aligns with the evolving regulatory landscape surrounding AI which emphasises "human-centred AI" and the holistic evaluation of human-AI team performance [31,32]. AI safety assessment is not a mere problem of computer science but also one of human-AI cooperation [33], which should incorporate behavioural elements grounded in human perceptual and decision-making studies.

We use the example of cardiovascular management in sepsis to study the behaviour of physicians in response to AI recommendations. Here, we share the results of an observational study of human-AI interaction in a high-fidelity simulation suite focusing on the influence of safe and unsafe AI recommendations on treatment decisions. We explore eye-tracking data and explainable AI questions deeper in a related/sister manuscript [34].

## Materials and methods

### Objective

We conducted an observational human-AI interaction study in a high-fidelity simulation facility. This study had two primary objectives. The first was to measure whether participants were able to detect, and correctly reject, unsafe recommendations from an AI tool (such as dramatically increasing doses of vasopressors to a patient with sepsis related to infected and leaky heart valves and elevated blood pressure) and/or ask for senior help when appropriate. The second was to identify whether or not human intervention could interfere with this interaction and sway clinicians away from their initial decision (potentially toward an unsafe treatment). Secondary study objectives included: (i) quantifying the shift in fluid and vasopressor doses induced by seeing an AI recommendation, and (ii) determining whether or not gaze patterns varied differentially depending on the safety status of the AI recommendation.

### Experimental design

Participants (clinicians) were briefed on the experiment and completed a pre-experiment questionnaire. The questionnaire recorded their demographics (age, gender, years of experience), prior experience with AI, as well as attitude towards AI through questions on their perception of AI benefits/issues. See S1 Appendix for the full content of the briefing and questionnaire. Participants were told that they would conduct a review of several adult patients with sepsis within a simulation suite (Imperial College Simulation Centre) and that they would need to prescribe appropriate doses of fluid and vasopressor for each patient both

before and after getting advice from an AI tool. Critically, physicians were told that the AI recommendation engine had been successfully validated in multiple retrospective settings but had not been prospectively evaluated. The simulation layout is shown in Fig 1a.

Each physician completed a total of six different patient scenarios, simulating a virtual "ward round". Each scenario started with physicians entering the simulation suite and conducting their assessment of the patient as they saw fit. Data sources within the room included a standard paper ICU bedside data chart with observations and blood results, an ICU handover note including details of the patient's presentation and medical history, a vital signs monitor and a physical patient mannequin (Simman 3G, Laerdal Medical, Stavanger, Norway) which could be examined (see S2 Appendix for the details of each patient scenario). A member of the research team played the role of the bedside ICU nurse who could only give standardised responses to any question, and was always the same person for all subjects. Following their assessment of the patient, physicians were asked to recommend a dose rate for fluids (ml/hr) and vasopressors (noradrenaline, mcg/kg/min) for the coming hour (to match the format of AI recommendations). Physicians rated their confidence on a 1–10 scale and whether or not they would like support for their decision from a senior physician (or a second opinion if the physician was

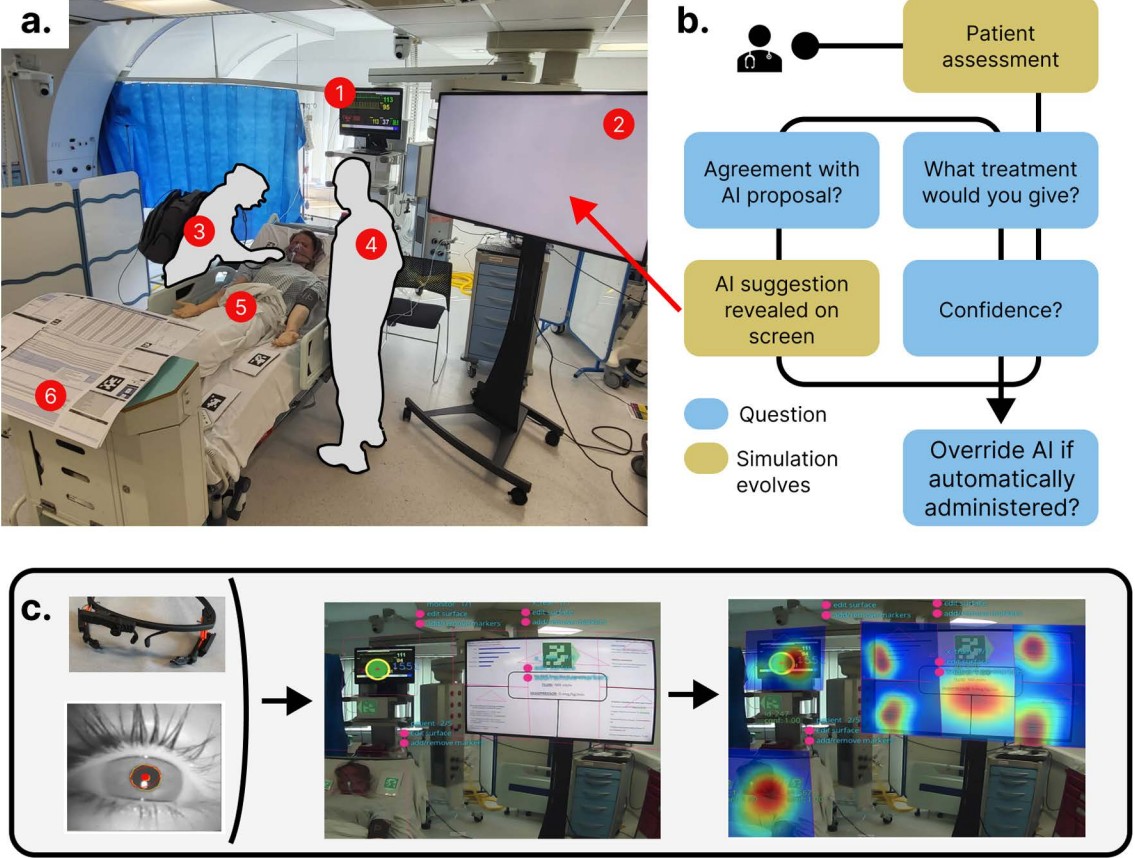

**Fig 1. Experimental design.** A. Visual representation of the simulation suite and overall summary of experimental design. A full description and rationale for the methods chosen including the realism limitation of the environment is given in the main text.: (1) Bedside monitor (2) AI screen (3) Participant (4) Bedside nurse (5) Patient mannequin (6) Intensive care unit (ICU) bedside information chart. B. Experimental protocol diagram. **C.** Gaze-based attention extraction pipeline: eye-tracking glasses, pupil camera view, a recorded field of view with April tags (QR codes) and reconstructed data with fixation heatmaps on the different regions of interest (ROIs).

already senior themselves). They were then shown the AI recommendation, asked to what extent they agreed with the recommendation on a 5-point Likert scale (from completely disagree to completely agree) [35], and then the initial dosing-related questions again (what dose they would prescribe, their confidence level, optional ask for senior help - see Fig 1b).

Finally, physicians were asked whether or not they would stop the AI recommendation if it were to be automatically administered to the patient. This question was intended to nuance the agreement prompt and identify situations where a participant might disagree with an AI recommendation but not necessarily consider it a threat to patient safety. Participants were clearly introduced to the nuance between these two questions in the pre-experiment briefing.

The running of a single patient scenario from entry into the simulation suite to exit constituted one trial. Trials were categorised by the nature of the AI tool's recommendation provided to the participant: safe, unsafe or "challenged" unsafe. In the latter, after the physician reported whether or not they would stop the AI recommendation if it were to be automatically administered, the bedside nurse was permitted three attempts (all following a standardised script) to verbally try to convince them to change their mind (see S3 Appendix for the scripts). The patient scenarios, including the AI recommendations, were piloted with specialist clinicians (not enrolled in this experiment) within the Imperial College Critical Care research group who validated their credibility. Each physician experienced four safe trials, one unsafe and one "challenged" unsafe in a pseudo-randomised order (see the trial matrix in S4 Appendix). The first trial encountered by every physician was always in the safe condition to establish a baseline level of trust with the AI tool and let the physician familiarise themselves with the environment. The number of safe and unsafe scenarios was chosen to balance recommendation type and statistical power. Physicians experienced more safe than unsafe trials to maintain some form of trust in the system. There was no time pressure on the physicians to make their decisions: in early dry-runs we estimated 7 minutes per trial, so we fitted 6 trials in the 45 minutes allocated to the experiment. Actual trial length depended on the speed of individual physicians. The details of each patient scenario are presented in S2 Appendix.

All AI recommendations were synthetically generated by the research team to ensure a standardised experimental format (i.e., they were not from a real AI system). The definition of unsafe recommendations was based on extreme under- or over-dosing of fluid and/or vasopressor as per previous work [13]. Our conservative approach to defining unsafe doses could lead to an overestimation of the stopping probability. Our design choice was justified by the clinical challenge of establishing dose-specific causality and the difficulty of recruiting large numbers of clinicians. All participating physicians were fully debriefed at the conclusion of the study on the synthetic nature of the AI recommendations so as not to bias their opinions of future interactions with AI-driven systems. Four types of explanations for the fictitious AI system were constructed:a natural language description of the Q-value difference between the recommended action and alternative actions (marker of the extent to which the optimal AI recommendation is significantly better than the alternative or only marginally better), the change in short-term mortality after dosing changes as predicted by the AI, the top-five ranked feature importance for input data contributing to the AI recommendation, and the three most influential training examples during the Q-learning process. These explanations are all realistic explanation types for reinforcement learning decision support systems [36]

During each trial, all physician responses were recorded by a member of the research team sitting in a dead angle in the simulation suite. This data, along with questionnaire answers was reformatted and analysed in Python (code available online here: https://figshare.com/s/78c5ff-5c6031f701c0d1). All statistical tests presented in this report compare proportions with a two-sided z-test, giving the probability of rejecting the hypothesis that the proportions of the compared groups are the same. When multiple comparisons are run on the same dataset, the

p-values are adjusted with the Bonferroni correction. When reporting proportions, we adopt the following format: "[mean] ± [standard error]%, 95% CI: [Weld confidence interval]". Any reference to a standard deviation is explicit.

## Eye-tracking for gaze recording

In this study, gaze was employed as an indicator of physicians' attentional focus during simulations, with particular interest in whether this varied according to the safety of the AI recommendation. Pupil and first-person videos were recorded with non-invasive commercially available eye-tracking glasses (Pupil Core headset). The Pupil Labs software (Core, version 3.3) utilised both eye cameras to delineate the pupil and estimate the direction of gaze within the recorded field of view.

Prior to the experiment, a two-part 2D calibration procedure was conducted. The initial stage involved a static calibration using five screen markers on a laptop display (default Pupil Labs 'screen marker' calibration). Subsequently, a depth-based static exercise was performed, requiring participants to focus on nine screen markers sequentially ('natural features' mode) displayed on a 60-inch TV screen, initially at 1 metre and then at 2-metre distance. A laptop (Lenovo Thinkpad) was connected to the eye-tracking glasses for the entire experiment. To allow for unrestricted movement in the suite, the glasses were connected via USB to a battery-powered laptop (Lenovo Thinkpad) worn by physicians in a lightweight backpack.

Subjects were eligible for eye-tracking data collection according to the following inclusion criteria: the Pupil Labs glasses were able to detect their pupils, the calibration step was passed successfully, and the glasses-laptop connection was not lost during the experiment. In total, 19/38 physicians passed the calibration exercises and had their gaze-based attention data collected. There was no evidence that this data loss led to any inclusion bias. 13/19 male doctors had gaze data (68% male compared to 66% male in the overall sample). 6/19 senior doctors had gaze data (32% senior compared to 55% senior in the overall sample). Physicians were instructed to point to where they were reading on the handover note at the start of each scenario as a final level of validation that the eye tracking was appropriately calibrated.

We defined four key regions of interest (ROIs) (Fig 1c): the paper ICU data chart, the vital signs monitor, the patient mannequin (Laerdal Simman 3G) and the AI display screen. Four further sub-regions were identified within the AI screen ROI corresponding to four types of explanation for the AI recommendation. April tags (simple QR codes) within the simulation suite (see Fig 1c) were used to identify ROIs in post-processing. As is common practice in eye-tracking literature [37,38], we used the number of gaze fixations per ROI—a fixation being the predominant eye movement occurring when the foveal region of the visual field is held stationary—as a proxy for participant attention.

## Participant recruitment and simulation facility

Recruitment of ICU physicians made use of both convenience sampling and targeted advertising to a local NHS trust (Imperial College Healthcare NHS Trust) Inclusion criteria were: (i) practising physician, (ii) has worked for two or more months in an adult ICU, (iii) currently works in ICU or has worked in ICU within the last 6 months. Physicians were compensated for their time and each experiment lasted approximately 60 minutes.

## Ethics statement

The study was approved by the Research Governance and Integrity Team (RGIT) at Imperial College London and the UK Health Research Authority (Ref: 22/HRA/1610). Every participant gave written consent to take part in the study.

## Results

A total of 38 intensive care physicians took part in the experiment (see Fig 2). This cohort comprised 25 men (66%) and 13 women (34%), proportions in line with the national population of intensivists [39]. The balance between junior and senior physicians (with less or more than 5 years of experience respectively) was even and 21% of participants reported having been personally involved or having had experience in AI research.

Each physician completed six (four safe, two unsafe) different patient scenarios leading to a total of 228 recorded trials. Of these trials, 76 featured an unsafe AI recommendation and 152 were safe ones. See S4 Appendix for the full trial matrix.

In total, unsafe AI recommendations were stopped more often than safe AI recommendations (83 ± 4%, 95% CI: 0.75–0.91 vs. 29 ± 4%, 95% CI: 0.21–0.37, p < 0.0001). The proportion stopping unsafe AI recommendations rose to 92 ± 3%, 95% CI: 0.86–0.98 (p = 0.0823) when including physicians who asked for a senior opinion, which would most likely lead to the unsafe AI recommendation being rejected (see Fig 3a). This analysis was further expanded by categorising physicians into junior (< 5 years of intensive care unit (ICU) experience) and senior (≥5 years of ICU experience) practitioners. There was a non-significant trend for junior physicians to stop AI recommendations less often than senior physicians (79 ± 7%, 95% CI: 0.65–0.93 vs. 83 ± 6%, 95% CI: 0.71–0.95, p = 0.11). Junior physicians asked more often for a second opinion than senior physicians (70 ± 4%, 95% CI: 0.62–0.78 vs. 28 ± 4%, 95% CI: 0.20–0.36, p < 0.0001), which led to more (though not significantly more) unsafe recommendations being stopped or escalated by juniors (94 ± 4%, 95% CI: 0.86–1, against 91 ± 5%, 95% CI: 0.81–1, by seniors, p = 0.12). To bring more depth to the analysis than a binary outcome, we investigated the extent of AI influence which was calculated on a continuous scale from 0 (completely ignoring advice) to 1 (completely relying on advice) using the formula [(final estimate – initial estimate)/ (advice – initial estimate)] per work by Yaniv et al. [40]. The safety status of AI recommendations did not significantly impact this metric, see S5 Appendix. Deeper analysis of AI influence on the same clinical task using the response shift paradigm can be found in earlier work [4,41].

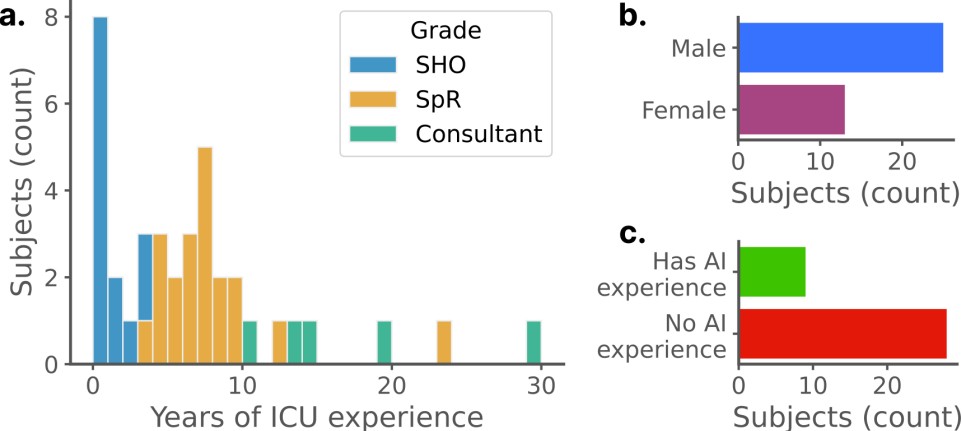

**Fig 2. Recruited cohort demographics (n = 38).** – A. Distribution of intensive care experience. Consultant, most senior and equivalent to attending in the United States (US); SpR, specialist registrar and equivalent to fellow in the US; SHO, senior house officer, most junior and equivalent to resident in the US. B. Gender distribution. C. Proportion of physicians who had ever been involved in a research project involving AI. This cohort covers the whole range of experience levels, is in line with national gender ratios and contains both people who have and have not worked with AI.

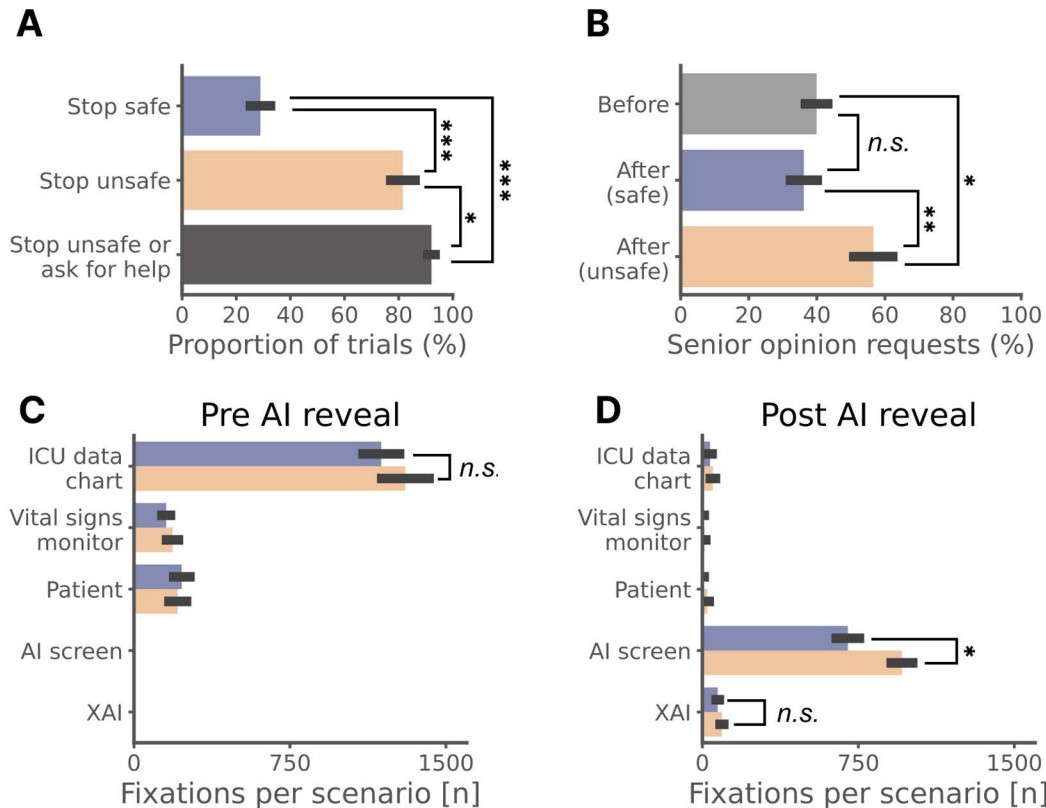

**Fig 3. Impact of AI recommendation safety status on clinician decisions, and gaze fixations on each ROI.** A. Bar chart of the proportion of stopped safe recommendations, stopped unsafe recommendations and stopped or escalated unsafe recommendations. B. Proportions of requests for senior help before seeing any recommendation and after having seen a safe or an unsafe one. C. Number of gaze fixations on each ROI before revealing the AI recommendation (i.e., there can be no fixations on the AI) D. Number of gaze fixations on each ROI after revealing the AI recommendation (when clinicians have already evaluated the non-AI information sources and so would be expected to look at these much less). Z-score Bonferroni-corrected proportion statistical tests: *** p < 0.0001, ** p < 0.005, * p < 0.05, *n.s.* not significant, error bars represent the standard error of the mean (SEM), XAI = explainable AI.

Similarly, second-opinion requests rose from 40 ± 3% (95% CI: 0.34–0.46) before seeing any AI recommendation to 57 ± 6% (95% CI: 0.45–0.69) after seeing an unsafe AI recommendation (p = 0.016) but the reduction in requests to 36 ± 4% (95% CI: 0.28–0.44) after seeing a safe AI recommendation was not significant (Fig 3b). Seeing an unsafe rather than a safe AI recommendation triggered more senior/second opinion requests (57 ± 6%, 95% CI: 0.45–0.69 vs. 36 ± 4%, 95% CI: 0.28–0.44, p = 0.0051). Seeing unsafe AI recommendations therefore significantly increased the proportion of requests for senior help.

As expected, prior to the AI recommendation being revealed, no significant difference in gaze fixations on regions of interest (ROIs) was observed between safe and unsafe scenarios regarding the three AI-independent regions (ICU data chart, vital signs monitor, and patient mannequin), see Fig 3c. Subsequent to the disclosure of the AI recommendation, there were more fixations on the AI screen in the unsafe scenarios (mean 960, SD 218) versus safe scenarios (mean 700, SD 236) (p = 0.0015, see Fig 3d). Finally, the number of gaze fixations on the AI explanation ROI was not significantly different between safe and unsafe scenarios.

The distributions of initial fluid and vasopressor dose prescriptions across participants in our six scenarios are shown in Fig 4. These results show wide variation in clinical practice,

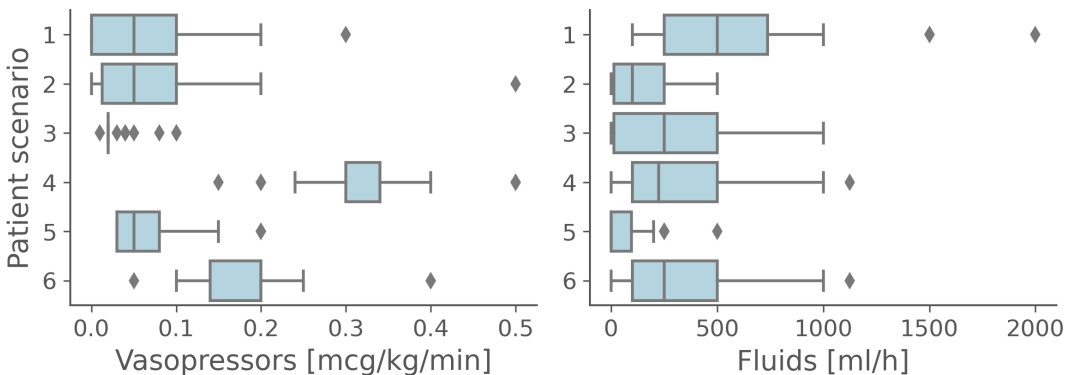

**Fig 4. Clinical practice variability.** Distribution of initial (i.e., pre AI reveal) vasopressor (left) and fluid (right) prescriptions by physicians for each patient scenario.

even when given the exact same information. Fig 4 suggests that the extent of the variation in prescribing might depend on case-specific features (e.g., in scenario 2, the patient had already had more fluids than in scenario 1 so physicians gave less fluid, or patient 5 had sepsis related to infected and leaky heart valves so physicians were more careful with both fluid and vasopressor). We also evaluated the extent to which being far from the human mean dose initially was indicative of not stopping unsafe AI recommendations. While subjects with most unsafe AI misses were further from the mean, there was no statistically significant correlation between the distance to human mean in initial dose and likelihood of accepting an unsafe AI recommendation (see S6 Appendix).

We also investigated the extent to which AI recommendations influenced prescription decisions. Physicians changed their prescription (dose of fluids and/or vasopressors) in 46 ± 3% (105/228), 95% CI: 0.40–0.52, of trials after seeing what the AI suggested. Both safe and unsafe AI recommendations influenced human decisions to different extents: fluid doses shifted on average by 80 ml/h (and vasopressor doses by 0.01 mcg/kg/min) after a safe AI recommendation compared to 40 ml/h (and 0.08 mcg/kg/min) after an unsafe AI recommendation. This difference can be attributed to the nature of unsafe AI suggestions, which involve under- or over-dosing of one or both drugs. The larger discrepancy between unsafe AI suggestions and physicians' typical practices likely contributes to the reduced influence of unsafe recommendations. Fig 5 shows the shift of distribution in vasopressor prescriptions before and after the AI recommendation was seen for two scenarios (split by whether the entire cohort is considered or only those physicians who did not ask for senior/second opinion). In scenarios (such as number two) where the unsafe AI recommendation was significantly influencing, this did not seem apparent when exclusively considering the physicians who did not request senior help (see S7 Appendix for this plot over all scenarios).

Finally, each physician had one of the two unsafe scenarios extended with a "challenge" section where the bedside nurse (a member of the experimental team) was given three attempts to change the physician's mind on whether or not to stop the AI recommendation were it to be automatically implemented. In 95 ± 4% (95% CI: 0.87–1.03) of cases (), the verbal input challenge from the bedside nurse did not change the decision of the physician to accept or reject the automated application of the AI recommendation. However, two participants (both junior) were persuaded to change their minds from interrupting an unsafe recommendation to accepting it. While we recorded previous experience with AI as an independent variable, our analyses did not find any impact on the primary outcomes of this study.

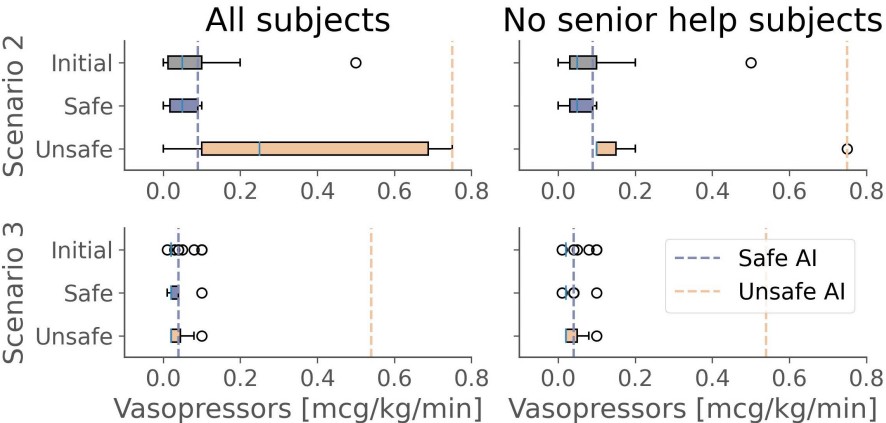

**Fig 5. Shift analysis.** Distributions of vasopressors dose distribution before and after having seen a safe or unsafe AI recommendation for two patient scenarios across all physicians (left) and only those who did not ask for senior help/second opinion (right). For each group of physicians and patient scenarios, we show the distributions of doses before seeing any recommendation (grey), after seeing a safe recommendation (blue), and after seeing an unsafe one (yellow). The dashed bars represent the values of the safe and unsafe AI recommendations for each patient scenario. Unsafe AI recommendations do not always influence the final decision. Physicians most influenced by unsafe AI recommendations tend to ask for a second opinion, while those who do not ask for help are less influenced by unsafe recommendations.

## Discussion

Our conclusions on the study objectives include the following:

- AI advice alters treatment decisions and therefore patient care;

- Physicians stop unsafe recommendations more frequently than safe ones, though not all unsafe AI advice was stopped;

- Junior physicians escalate their decision more often than senior physicians do, and unsafe AI recommendations lead to more decision escalation than safe ones;

- Unsafe AI recommendations drew more direct attention than safe ones but the visual attention brought to AI explanations did not differ;

- Intervention from human colleagues can sway a minority of physicians to follow an unsafe recommendation.

These findings show the importance of educating clinical teams who will interact with a new AI recommender system on the correct intended use of the system, including target patient population, indications, and limitations, as well as the importance of clinical context when integrating the AI recommendations into their practice. We hypothesise that the human influence towards unsafe recommendations could have been avoided by a small briefing on the key points touched on by the bedside nurse: technology benefits, limitations, and legal paradigms. Domain experts should explore how to provide effective education given the context of limited time, resources, and varying AI literacy among clinicians. Similarly, high-level supervision of junior clinicians using these tools would ensure constant availability of a second opinion to discuss the system's output in cases where there might be disagreement.

This study reinforces the call for more interdisciplinary and realistic human-AI interaction studies on domain experts [42,43]. Critically, our experimental design also allowed us to study human-human behavioural dynamics during an encounter with AI decision support. This is

important as most clinical uses of AI-driven decision support tools will be in the context of multi-disciplinary teams where humans other than the final decision-maker can still positively or negatively influence the interaction between the final human decision-maker and the AI. This is why our study was run in a simulation suite: an environment that reproduces natural stimuli of bedside practice for clinicians without any risk to patients.

Our findings highlight important implications for clinical practice that warrant further exploration. Recent work by Hasan et al. examines the ethical considerations of AI in pharmacy practice and reveals similar themes around patient autonomy, data protection, and the need for regulatory frameworks [44]. Their study found widespread concerns about patient data privacy, cybersecurity threats, and lack of legal regulation among pharmacy professionals. To address these issues in clinical settings, we recommend developing comprehensive education and training programs for healthcare professionals that encompass both technical skills and ethical considerations. Such programs could include AI literacy modules, case studies on responsible AI integration, ethical frameworks for evaluating AI recommendations, and hands-on training with AI tools. Healthcare institutions should also consider establishing AI ethics committees to provide ongoing guidance and oversight.

The ethical implications of using AI in clinical decision-making are profound. While AI has the potential to enhance diagnostic accuracy and treatment optimization, it also raises concerns about the erosion of human judgement and the patient-provider relationship. To mitigate these risks, we advocate for a human-centred approach to AI integration that preserves clinician autonomy while leveraging AI as a supportive tool. By proactively addressing these ethical and practical challenges, healthcare systems can work towards responsible AI adoption that enhances rather than compromises the quality and humanity of patient care.

While eye-tracking is typically used in controlled environments [45], this study demonstrates the feasibility of using this behavioural phenotypic marker of attention in more realistic, less constrained, environments. Our findings indicate that physicians fixated more on unsafe than safe AI recommendations implying an appropriately higher level of allocated cognitive attention. However, we also observed that physicians did not rely more on AI explanations in the unsafe scenarios calling into question the use of explanations as a mitigation strategy for unsafe AI. Nor did physicians devote significantly more attention to looking back at the 'traditional' (non-AI based) clinical data after seeing an unsafe AI recommendation to understand why the recommendation might be unsafe (i.e., there was no outward evidence of a desire to 'debug' the unsafe AI recommendation). The eye-tracking and explainability aspects of this experiment are discussed in greater depth in a sister manuscript [34].

The influence of AI recommendations on clinical judgement has been studied previously in vignette-type experiments. In tasks with a gold standard to compare against like medical imaging, AI advice has been shown to impact clinical decisions differently depending on its quality and the level of expertise of the clinical user. In this context, explainability did not help physicians identify biased models [46,47]. Similarly, even in settings with no gold standard like cardiovascular management of sepsis in intensive care, AI advice has been shown to significantly shift final treatment decisions in a vignette setting [4]. This work goes one step closer to clinical deployment by studying these interactions in a high-fidelity simulation environment. This setup enabled the study of human-AI interaction with eye-tracking as well as the ability to investigate human-human interactions as they relate to AI. Most safety studies of clinical decision support systems use medication error as both the primary outcome measure and proxy for patient safety [26,48]. Here, we look at systems that are not yet deployed in clinical practice, so measuring prescription error rate directly (and correlating that to 'error' without a gold standard) is challenging. Therefore, we tackled the problem from a different

angle and aimed to estimate the ability of clinicians to spot unsafe treatment recommendations from an AI tool.

The limitations of the study should also be acknowledged. First, prescribing hourly fluid and vasopressor doses directly is unusual for intensive care physicians who typically indicate blood pressure or urine output targets and let the bedside nurse titrate the actual doses within a reasonable range to reach the set targets. However, direct drug dose recommendation has become the standard for reinforcement learning (RL)-based decision support tools in healthcare [49,50]. While clinicians do not usually work directly with drug doses in their current practice, the AI systems they will encounter in their practice likely will and so our study better replicates the actual context in which clinicians will interact with AI prescription decision support tools in the ICU. Similarly, the simulation limited the physician's action space to one specific aspect of patient care, preventing action plans that might go beyond the defined possibilities (e.g., altering sedation levels and antimicrobial choices). Moreover, making treatment decisions for the next hour is less dynamic than real clinical practice (where for example the ability to examine a real patient and use advanced cardiac output monitors might add to the nuance of the overall clinical picture).

Although the simulation suite replicates many of the features of the clinical environment (including real bedside monitors displaying data in real time, with working alarms and some human interactions) it cannot completely replicate the competing priorities and time-pressured clinical environment of a real intensive care unit. Additionally, we were not able to replicate the dynamic sequential assessment/ treatment decisions provided over time, as we asked subjects for a single decision on each of six separate patients. Assessing multiple decision points in each scenario would have been more reflective of real clinical practice, but would have created an exponential number of potential scenarios, and therefore led to less similar situations to compare in the statistical analysis.

While the data is not the same as might be obtained from real-world settings, it is nonetheless far higher fidelity than comparable vignette studies. Moreover, while running the experiment in a simulation suite increases the ecological validity of the results, it also makes recruitment harder as physicians must physically attend (compared to vignettes which can be performed remotely over the internet). In terms of other data collection inefficiencies, eye-tracking technology fails to capture pupil position for some subjects thus limiting the statistical power of gaze analysis. This study was run in one simulation suite in a modern university hospital which, although providing a controlled and realistic setting, may not fully capture the diversity and nuances present in different hospital environments. Potential biases from the use of convenience sampling were not particularly mitigated against, we thought it was a good trade-off between clinicians' low availability (frequent out of hours shifts and heavy clinical commitments) and the study's recruitment target.

Finally, the results from this study demonstrate the existence of human-human influence, within the context of interacting with AI, but the sample size limits the capacity to accurately estimate the likelihood and significance of the phenomenon or generalise it to different settings. The human-human influence in a human-AI team remains an open research field, especially for multi-disciplinary teams.

From a different perspective, one could challenge the definitions of safe and unsafe recommendations used in the scenarios by arguing that there is no ground truth in sepsis resuscitation and that they are therefore subjective. One might even go further and argue that strategies that under- or over-dose in specific patients (compared to the 'average') could be desirable in some cases. The scenarios used in this experiment were designed for the unsafe recommendations to be inappropriate to a majority of clinicians and validated by an independent panel of intensivists. The introduction of AI-driven decision support tools, particularly

those using reinforcement learning, aims to improve patient outcomes beyond the current standard of care [51,52]. This means that such systems will give recommendations that differ from what the clinical team would ordinarily have done but potentially without explanation - a "mysterious oracle dilemma" where the AI oracle recommends actions that on average lead to better outcomes but might occasionally be suboptimal, and the users do not get context on the AI recommendation. It will therefore be essential for humans to exert critical thinking and assess how reasonable the AI recommendation is to filter potentially novel but superior calls by the AI from harmful recommendations. This dilemma relates to philosophical discussions on epistemic authority and testimonial knowledge [53,54], and deserves more discussion in future works. Critically, this study does not claim to control for all the potential failure modes of AI recommendation systems in the ICU. Questions of connectivity, data mapping and updates should also be addressed before deploying such systems. For researchers seeking to evaluate safe and unsafe recommendations similarly, we recommend grounding the choice of criteria for decision safety in a combination of legal documentation (i.e., which decision could make the user liable), and/or opinions from a panel of domain experts (in this case, experienced intensivists) as done in (1).

Another interesting angle for analysis is cognitive load, which can be evaluated through eye-tracking proxies [55,56]. These metrics did not show significant correlations with estimated scenario difficulty (see S8 Appendix). Fig 3 shows that recommendation safety impacted the number of fixations on the screen, indicative of a higher mental load. The question of task load on clinicians in a real context with AI interaction should be explored further with tools like NASA TLX [57], which are however subjective questionnaires.

As regulators require clear intended purpose statements for software as medical devices [58], our high-fidelity eye-tracking-based approach to evaluating an AI-driven decision support tool serves as a basis for promoting the generation of safety evidence, with a particular emphasis on ecologically valid settings. This work also provides pointers on how medical education could tailor for the increasing use of AI technologies in medical decision-making. Furthermore, the recent rise in popularity of generative AI (most notably as large language models) highlights the safety concerns of hallucinatory outputs that might be acted upon in a clinical setting and bring harm to patients [59,60]. The unsafe AI recommendations in this work share characteristics with generative AI hallucinations: they are presented to users with no warning and as much confidence as safe recommendations. It is likely that an AI system that shows overall super-human performance in a given task will still show lower-quality performance in some specific cases [61]. Solutions such as uncertainty-aware models or explainable AI might help users differentiate between well-informed recommendations and flawed calls [41,62,63]. The human-AI interactions at the bedside, with a particular focus on high-pressure decision-making, would also help to accelerate the safe translation of AI-based decision support tools to the bedside.

To conclude, it is critical for clinician acceptance, regulatory compliance and real-world adoption that we evaluate cooperation between clinical experts and AI decision-support tools in high-fidelity settings. This study demonstrates the influence of AI recommendations on clinical behaviour and suggests that the vast majority of unsafe AI recommendations are appropriately rejected by bedside clinicians. The findings on junior physicians occasionally accepting an unsafe AI recommendation and their general willingness to seek senior help when unsure should inform the intended use (i.e., some tools might need to be only used by junior clinicians if they have access to senior advice). We study the role of XAI in greater detail in our sister manuscript [64]. Uncertainty awareness, novel forms of AI interpretability and a better understanding of human-human interactions (i.e., team decisions) in the context of AI-driven decision support will help not only with assuring safety from a regulatory perspective but also in fostering confidence and trust from clinician end-users.

## Supporting information

**S1 Appendix.** Pre-experiment questionnaire filled in by every subject to draw a picture of each subject's profile.
(DOCX)

**S2 Appendix.** List of all patient scenarios used in this study, with their safe and unsafe versions, as well as a justification for them.
(DOCX)

**S3 Appendix.** Scripts of the challenges used by the accomplice bedside nurse to challenge toe clinician's decisions to follow or not the AI in the "challenged unsafe" arm of the experiment.
(DOCX)

**S4 Appendix.** Trial matrix of the experiment showing the psrudo-random order of trials and conditions.
(DOCX)

**S5 Appendix.** Extension of the persuasion coefficient analysis of both safe and unsafe AI recommendations.
(DOCX)

**S6 Appendix.** Analysis of the link between clinicians giving initial doses far away from the average of their peers and their likelihood of accepting an unsafe AI recommendation.
(DOCX)

**S7 Appendix.** Extension of Fig 5 showing the dose distribution shifts for all patient scenarios and conditions.
(DOCX)

**S8 Appendix.** Graphs of eye-tracking based cognitive load metrics.
(DOCX)

## Author contributions

**Conceptualization:** Paul Festor, Myura Nagendran, Anthony C. Gordon, Aldo A. Faisal, Matthieu Komorowski.

**Data curation:** Paul Festor, Myura Nagendran.

**Formal analysis:** Paul Festor, Myura Nagendran, Aldo A. Faisal.

**Funding acquisition:** Anthony C. Gordon, Aldo A. Faisal.

**Investigation:** Paul Festor, Myura Nagendran, Aldo A. Faisal, Matthieu Komorowski.

**Methodology:** Paul Festor, Myura Nagendran, Aldo A. Faisal.

**Project administration:** Anthony C. Gordon, Aldo A. Faisal, Matthieu Komorowski.

**Resources:** Aldo A. Faisal.

**Software:** Paul Festor, Myura Nagendran, Aldo A. Faisal.

**Supervision:** Anthony C. Gordon, Aldo A. Faisal, Matthieu Komorowski.

**Validation:** Myura Nagendran.

**Visualization:** Paul Festor, Myura Nagendran.

**Writing – original draft:** Paul Festor, Myura Nagendran.

**Writing – review & editing:** Paul Festor, Myura Nagendran, Anthony C. Gordon, Aldo A. Faisal, Matthieu Komorowski.

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
