## [Decision Letter · Decision Letter 0]

13 Jun 2024

PDIG-D-24-00200

(Mis)leading the doctor: safety of human-AI cooperative decision-making in hospital settings

PLOS Digital Health

Dear Dr. Faisal,

Thank you for submitting your manuscript to PLOS Digital Health. After careful consideration, we feel that it has merit but does not fully meet PLOS Digital Health's publication criteria as it currently stands. Therefore, we invite you to submit a revised version of the manuscript that addresses the points raised during the review process.

Please submit your revised manuscript within 60 days Aug 12 2024 11:59PM. If you will need more time than this to complete your revisions, please reply to this message or contact the journal office at digitalhealth@plos.org. Please include the following items when submitting your revised manuscript:

We look forward to receiving your revised manuscript.

Kind regards,

Akhilanand Chaurasia

Section Editor

PLOS Digital Health

Journal Requirements:

1. Please send a completed 'Competing Interests' statement, including any COIs declared by your co-authors. If you have no competing interests to declare, please state "The authors have declared that no competing interests exist". Otherwise please declare all competing interests beginning with the statement "I have read the journal's policy and the authors of this manuscript have the following competing interests:"

2. Please ensure that Funding Information and Financial Disclosure Statement are matched.

3. In the Funding Information you indicated that no funding was received. Please revise the Funding Information field to reflect funding received.

4. We ask that a manuscript source file is provided at Revision. Please upload your manuscript file as a .doc, .docx, .rtf or .tex.

5. Please provide separate figure files in .tif or .eps format only and remove any figures embedded in your manuscript file. Please also ensure that all files are under our size limit of 10MB.

Additional Editor Comments (if provided):

Reviewers' comments:

Reviewer's Responses to Questions

**Comments to the Author**

1. Does this manuscript meet PLOS Digital Health’s publication criteria ? Is the manuscript technically sound, and do the data support the conclusions? The manuscript must describe methodologically and ethically rigorous research with conclusions that are appropriately drawn based on the data presented.

Reviewer #1: Yes

Reviewer #2: Partly

Reviewer #3: Partly

Reviewer #4: Yes

2. Has the statistical analysis been performed appropriately and rigorously?

Reviewer #1: Yes

Reviewer #2: Yes

Reviewer #3: I don't know

Reviewer #4: Yes

3. Have the authors made all data underlying the findings in their manuscript fully available (please refer to the Data Availability Statement at the start of the manuscript PDF file)?

Reviewer #1: Yes

Reviewer #2: Yes

Reviewer #3: Yes

Reviewer #4: Yes

4. Is the manuscript presented in an intelligible fashion and written in standard English?

PLOS Digital Health does not copyedit accepted manuscripts, so the language in submitted articles must be clear, correct, and unambiguous. Any typographical or grammatical errors should be corrected at revision, so please note any specific errors here.

Reviewer #1: Yes

Reviewer #2: No

Reviewer #3: No

Reviewer #4: Yes

5. Review Comments to the Author

Please use the space provided to explain your answers to the questions above. You may also include additional comments for the author, including concerns about dual publication, research ethics, or publication ethics. (Please upload your review as an attachment if it exceeds 20,000 characters)

Reviewer #1: The research on AI and human interactions in acute healthcare settings is highly valuable, enhancing our understanding of how AI-based clinical decision systems impact healthcare workers and physicians’ decision-making processes. The detailed description of the work, along with the comprehensive appendices, greatly facilitates the reproducibility and appraisal of the research by other scholars.

However, the layout of the manuscript could benefit from further refinement. I recommend that the authors have the manuscript reviewed by proof readers not involved in the work to ensure that the text flows coherently for the average reader.

Additionally, the recommendations and lessons drawn from the study should be clarified and distinctly separated from the discussion section. Including a flow chart or a recommendation box could effectively address this issue.

Reviewer #2: Dear Editor-in-Chief, Dear Authors,

I am writing to provide my review of the manuscript titled "(Mis)leading the doctor: safety of human-AI cooperative decision-making in hospital settings" authored by Festor et al., which was submitted for consideration in PLOS Digital Health. I have thoroughly reviewed the manuscript, and I am pleased to share my insights and evaluation with you.

I would like to thank the authors for their submission. Trying to understand the human-AI interaction, especially in such a situations like ICU environment is very critical for the efficacy and safety of healthcare. 

In general, I still find the article somewhat confusing, mainly due to a bit off order of chapters. It would be important for instance to read the methods before the results to really understand what this is all about. Also, I would suggest authors to introduce the human decision limitations in the introduction. Otherwise criticizing human-AI interaction is only the other side of the coin. 

Also, I am a bit confused why the least prioritized author is the corresponding author here.

Here are my observations:

Title: 

I would suggest the title to be less populistic and more scientific. 

General: 

I suggest you use this order for the headings: 

• Abstract

• Introduction

• Materials and Methods

• Results

• Discussion

• Conclusions

Comments:

113: “However, the final decision remains in human hands.” – not always, I would argue Medtronic insulin pump with AI algorithm running the show independently is a high-stake decision making on whether to inject insulin or not. The word “usually” should be added, and the “high-stakes settings” should be defined.

115-116: What are the key differences?

117: …requirements… such as?

117: Grammar 

124: Grammar

125: What do you mean by limited availability of human experts?

126-128: You admit that the optimal care is unclear, yet your title and the whole article seems to warn that Human-AI interaction is not perfect. This is confusing, what is perfect and clear then?

123-131: I would add something about the bias where the physicians in unclear high-stakes settings do though trust more on their colleagues, who to be frank are also chaotic, random, prone to mistakes, and are making decisions as “black box”. Comparison to standard process is needed in my opinion in the introduction. Also, integrate this part with 140-141.

143: ecologically? What do you mean?

162-175 this belongs to Materials and Methods, not here, and can be ignored once the order is correct

180: grammar

182: which were safe / unsafe?

255: surely “confirm” is too strong word here?

274: I understood quite major part of the subjects failed to get their eye tracking data collected? That’s a limitation

301: RL-based?

395: Likert scale data analysis?

426-428: this belongs to discussion not here

433-438: no verbs => make a bullet points or a list of these 4 explanations

Results: 

It would great to summarize results of primary and secondary objectives, also in abstract

Other: 

Limitations of “physical simulation suite” vs real world testing? The data is surely not same as from the real world settings. 

Also, study was rather small so it is a limitation. Also, it was performed only in one(?) ICU in UK so no variation between different ICUs and/or countries / cultures can be explored. 

Figures:

Fig1 make all A/B/C letters either capital or not, now both

Fig1: B. What does “Override AI if automatically administered?” mean? I am not sure whether automatically means = before asking AI or similar?

Fig1: what is “April”?

Fig2 A. What are SHO and SpR?

Fig3. What is XAI?

Appendix S1: 

check grammar (e.g. one parenthesis is missing)

Appendix S6: 

“Cp” should be introduced in the text. The equation and then the explanation contains both capital and small letters, please unify. Also please check the grammar.

Are there only two colors in the bars, as explained in the label? It seems there are multiple colors used.

Reviewer #3: The title uses the term "(Mis)leading," which can be ambiguous. Also, the title does not specify the context clearly. Mentioning that this study is within "hospital settings" is broad; specifying "intensive care units" would provide more precise context.

The abstract can be refined for greater clarity, precision, and impact.

Lines 112-121: The introduction could benefit from a clearer definition of what constitutes "safe" and "unsafe" AI recommendations earlier in the section. Providing examples or a brief explanation here would help contextualize the study for readers. Also, it lacks depth in discussing previous research specifically focused on AI safety in medical decision-making.

Lines 122-131: While the example of cardiovascular management during sepsis is relevant, the rationale for choosing this specific example over others in healthcare is not well justified. Additionally, the discussion about the slow translation to bedside due to patient safety concerns (line 131) would benefit from more concrete examples or data to support this claim.

Lines 156-160: The explanation of the observational study is clear. Nonetheless, it would be beneficial to include more information on how the AI recommendations were generated and validated.

Lines 161-165: The study's primary aim is clear. However, the secondary objectives, particularly the quantification of shifts in doses and gaze patterns, seem less central to the main safety concern.

Lines 186-190: You must include more statistical details, such as confidence intervals for the proportions reported, to give a clearer picture of the variability in the data. The statistical significance of the results (p<0.0001) is strong, but the discussion on the non-significant trend for junior vs. senior physicians (lines 191-196) is confusing. It would be beneficial to clarify what these non-significant trends imply and how they might still be relevant. Including qualitative data or direct quotes from participants could provide deeper insights.

Lines 198-201: The mention of the persuasion coefficient from social sciences (line 198) is underexplained.

Lines 202-208: The findings on the increase in senior/second opinion requests are insightful, but the connection to overall AI safety could be better articulated. How does this behavior impact patient outcomes or decision-making quality?

Lines 210-217: The use of gaze fixation data (lines 212-217) is innovative, but the implications of these findings are not fully explored. How do these gaze patterns correlate with decision accuracy or safety?

Lines 228-238: The analysis of AI recommendation influence on prescriptions (lines 228-238) would benefit from a more detailed discussion on why safe recommendations had a different influence compared to unsafe ones. What factors contributed to these differences?

Lines 240-248: The "challenge" section with the bedside nurse (lines 240-248) provides valuable insights into human-to-human interactions influencing AI decisions. However, the sample size (only two participants swayed) is too small to draw any robust conclusions. This limitation should be acknowledged more explicitly.

Lines 255-263: The discussion would benefit from a more in-depth exploration of the implications of these findings for clinical practice. For example, what specific training or support might help clinicians better evaluate AI recommendations? Engage more thoroughly with existing literature. For example, discussing Hasan et al. (2024) would contextualize your findings within the broader field of ethical considerations in healthcare practice. https://doi.org/10.1186/s12910-024-01062-8

Integrate a section discussing the ethical implications of using AI in clinical decision-making, drawing parallels with the concerns raised in the pharmacy practice study. Recommend enhanced education and training for healthcare professionals to improve their understanding of AI, addressing both ethical concerns and practical challenges.

Lines 255-256: The assertion that AI recommendations "can influence clinician behaviour" is overly simplistic. This is not a novel insight, as the influence of AI on decision-making is a well-documented phenomenon across various fields. The discussion would benefit from a more nuanced exploration of how different types of AI recommendations influence clinician behavior in specific contexts.

Lines 258-263: The emphasis on the need for educating clinical teams is appropriate but not sufficiently developed. The statement is vague about the specifics of the educational content and methods. Moreover, the discussion should address the potential challenges and barriers to effective education, such as time constraints, resource availability, and varying levels of AI literacy among clinicians.

Lines 266-270: The claim that the study's design allows for the observation of human-human dynamics in AI interaction lacks empirical support. The authors should provide more detailed evidence from their data on how these dynamics manifest and their impact on decision-making. Additionally, more clarity is needed on how these observations can be generalized beyond the study's simulated environment.

Lines 271-273: The authors should discuss the limitations of simulation suite approach more thoroughly. Simulation environments cannot fully replicate the complexities and pressures of real clinical settings. The discussion should address the potential biases introduced by the simulated environment and how they might affect the study's findings.

Lines 276-283: The manuscript could be strengthened by discussing the limitations of eye-tracking as a measure of cognitive load and suggesting complementary methods that could be used in future research. The authors should explore alternative explanations for this behavior, such as inherent human attention biases toward negative stimuli. Additionally, the practical implications of these findings for clinical practice and AI system design are not adequately addressed. Also, the authors should delve deeper into why this might be the case. Possible factors could include mistrust in AI explanations, lack of clarity in the explanations provided, or cognitive overload. The discussion should propose potential solutions or areas for further research to enhance the usability and trustworthiness of AI explanations.

Lines 296-309: It would be helpful to discuss potential strategies to address these limitations in future studies. Also, prescribing hourly fluid and vasopressor doses is not typical practice for intensive care physicians and raises concerns about the ecological validity of the study.

Lines 311-331: The critique of the definitions of safe and unsafe recommendations is crucial. The discussion stops short of proposing concrete ways to address this issue. The authors should consider suggesting standardized guidelines or frameworks for defining and evaluating AI recommendations in clinical settings. Furthermore, the concept of the "mysterious oracle dilemma" is intriguing and warrants deeper exploration. How can clinicians balance the potential benefits of novel AI recommendations with the risks of occasional suboptimal outcomes?

Lines 333-342: The authors should provide more specific recommendations for how their findings can inform regulatory practices. Additionally, the potential role of generative AI in clinical settings is briefly mentioned. This is a critical area that deserves more attention, particularly in light of recent advances in AI technology.

Lines 346-357: In conclusion consider adding a forward-looking statement on the next steps for research in this area and how the findings could influence policy and practice in AI deployment in healthcare.

Lines 360-372: The study design briefing and pre-experiment questionnaire details (line 371) are relegated to the supplementary material. Including a summary of key questions and their relevance to the study will strengthen this section.

Lines 379-418: In the description of the simulation scenarios (lines 379-418), the justification for the specific number of scenarios (six) and the mix of safe, unsafe, and "challenged" unsafe trials could be better explained. Why was this mix chosen, and how does it ensure robust data collection?

Lines 421-431: The synthetic generation of AI recommendations (lines 421-431) ensures standardization but also raises concerns about ecological validity. How realistic were these recommendations perceived by the participants, and how might this affect the study's conclusions?

Lines 451-485: In the use of eye-tracking technology (lines 451-485), the high exclusion rate (only 19/38 physicians) due to calibration issues is problematic. This could introduce bias, and the impact of this exclusion should be discussed more thoroughly.

Lines 487-492: The recruitment strategy (lines 487-492) seems sound, but the use of convenience sampling could lead to selection bias. More details on how this was mitigated or its potential impact on the findings would be beneficial.

Technical Corrections:

Line 117: "understanding" should be capitalized as it starts a new sentence.

Line 120: Replace a comma after "yet" for better readability.

Lines 144-147: The word "simulation" is repeated; consider rephrasing for clarity.

Line 244: "bedside nurse did not sway the physician's decision" - consider rephrasing.

Figures and Tables:

Ensure all figures and tables are clearly labeled and referenced in the text. The legends should provide sufficient detail to understand the content without referring back to the main text.

Fig 1: The experimental design lacks clarity in conveying the simulation's realism and its applicability to real-world settings. The choice of using a mannequin and eye-tracking technology does not sufficiently address potential limitations such as the lack of dynamic interaction with a real patient and the artificiality of the environment, which could influence clinician behavior differently compared to a real ICU setting.

Fig 2: The sample size is not mentioned, which is critical for assessing the study's statistical power and the generalizability of the findings. Additionally, the proportion of participants previously involved in AI research is noted, but the potential impact of their prior experience on study outcomes is not explored.

Fig 3: The interpretation is overly simplistic. The discussion should provide a more thorough analysis of why clinicians might stop safe recommendations or escalate unsafe ones. The statistical significance (p-values) needs more context to understand the practical relevance of these findings.

Fig 4: The observed variability in clinical practice underscores the complexity of clinical decision-making. However, the figure fails to adequately correlate these variances with AI influence, and the discussion does not sufficiently address how AI might standardize or exacerbate these differences.

Fig 5: The conclusion that unsafe AI recommendations do not always influence the final decision needs more robust evidence. The discussion should delve deeper into the factors that contribute to a physician's decision to seek a second opinion and how this behavior interacts with AI recommendations.

Appendices

The questionnaire lacks depth in exploring potential biases or preconceptions that could influence study outcomes.

The nurse challenge scripts does not adequately analyze how these challenges affected clinician decision-making and their interactions with AI recommendations.

The detailed patient scenarios are comprehensive, yet the variability in clinical presentations might introduce uncontrolled variables. The justification for AI actions needs a more rigorous examination to ensure they are perceived as credible by the participants.

The use of persuasion coefficients requires a more detailed explanation and justification. The lack of significant differences between safe and unsafe persuasion distributions should prompt a reevaluation of the metric's sensitivity and applicability.

The attempt to correlate eye-tracking metrics with cognitive load and case difficulty lacks a significant findings suggests either a limitation in the measurement approach or the need for a more refined analysis.

Reviewer #4: The manuscript presents a study on the interaction between human clinicians and artificial intelligence (AI) decision-support systems in a high-fidelity simulation environment. The study aimed to evaluate the ability of clinicians to detect and reject unsafe AI recommendations in a sepsis management scenario.

The study involved 38 intensive care physicians who participated in a simulated ward round, where they were presented with six patient scenarios and asked to prescribe fluid and vasopressor doses before and after receiving AI recommendations. The AI recommendations were synthetically generated and included safe, unsafe, and "challenged" unsafe scenarios.

I really enjoyed reading the manuscript as its main messages are extremely important for our understanding of implementing AI tools in clinical practice. I only have a few minor suggestions:

1. The authors state that Clinicians did not look back at the original data to "debug" why an unsafe recommendation was given by the AI system. It might be interesting to understand, why clinicians didn't try to debug. Were the participating clinicians interviewed and confronted with this finding afterwards?

2. The sentence in line 162 is redundant and could be removed.

3. It seems that the majority of clinicians were male. Did the authors investigate, if gender may have an impact on the results?

4. From a statistical point of view, the authors used a relatively low number of clinicians/subjects in their study, which is appropriate for a pilot study. What remains unclear to me is, how the number of participating clinicians was estimated, if at all (the authors state in the manuscript that "The number of safe and unsafe scenarios was chosen to balance balance recommendation type and statistical power").

6. PLOS authors have the option to publish the peer review history of their article (what does this mean? ). If published, this will include your full peer review and any attached files.

**Do you want your identity to be public for this peer review?** For information about this choice, including consent withdrawal, please see our Privacy Policy .

Reviewer #1: Yes: Dr Zaki Almallah MD Clinical Professor

Reviewer #2: No

Reviewer #3: No

Reviewer #4: Yes: Ludwig Christian Hinske

---

## [Decision Letter · Decision Letter 1]

17 Dec 2024

Safety of human-AI cooperative decision-making within intensive care: a physical simulation study

PDIG-D-24-00200R1

Dear Prof. Dr. Faisal,

We are pleased to inform you that your manuscript 'Safety of human-AI cooperative decision-making within intensive care: a physical simulation study' has been provisionally accepted for publication in PLOS Digital Health.

Best regards,

Martin G Frasch

Section Editor

PLOS Digital Health

**Additional Editor Comments (if provided):**

**Reviewer Comments (if any, and for reference):**

Reviewer's Responses to Questions

**Comments to the Author**

1. If the authors have adequately addressed your comments raised in a previous round of review and you feel that this manuscript is now acceptable for publication, you may indicate that here to bypass the “Comments to the Author” section, enter your conflict of interest statement in the “Confidential to Editor” section, and submit your "Accept" recommendation.

Reviewer #2: All comments have been addressed

Reviewer #3: All comments have been addressed

2. Does this manuscript meet PLOS Digital Health’s publication criteria ? Is the manuscript technically sound, and do the data support the conclusions? The manuscript must describe methodologically and ethically rigorous research with conclusions that are appropriately drawn based on the data presented.

Reviewer #2: Yes

Reviewer #3: Yes

3. Has the statistical analysis been performed appropriately and rigorously?

Reviewer #2: Yes

Reviewer #3: I don't know

4. Have the authors made all data underlying the findings in their manuscript fully available (please refer to the Data Availability Statement at the start of the manuscript PDF file)?

Reviewer #2: Yes

Reviewer #3: No

5. Is the manuscript presented in an intelligible fashion and written in standard English?

Reviewer #2: Yes

Reviewer #3: Yes

6. Review Comments to the Author

Reviewer #2: (No Response)

Reviewer #3: Overall, the authors have responded thoughtfully to critiques, adding depth and clarity to their manuscript.

7. PLOS authors have the option to publish the peer review history of their article (what does this mean? ). If published, this will include your full peer review and any attached files.

**Do you want your identity to be public for this peer review?** For information about this choice, including consent withdrawal, please see our Privacy Policy .

Reviewer #2: No

Reviewer #3: No
